# Cannabinoids Activate Endoplasmic Reticulum Stress Response and Promote the Death of Avian Retinal Müller Cells in Culture

**DOI:** 10.3390/brainsci15030291

**Published:** 2025-03-10

**Authors:** Ana Lúcia Marques Ventura, Thayane Martins Silva, Guilherme Rapozeiro França

**Affiliations:** 1Neuroscience Program, Department of Neurobiology, Federal Fluminense University, Niterói CEP 24210-201, RJ, Brazil; thayanems@hotmail.com; 2Department of Physiological Sciences, Federal University of the State of Rio de Janeiro, Rio de Janeiro CEP 20211-040, RJ, Brazil; guilherme.franca@unirio.br

**Keywords:** cannabinoids, retina, Müller glia, oxidative stress, ER stress, apoptosis

## Abstract

Background/Objectives: Activation of cannabinoid CB1 or CB2 receptors induces the death of glial progenitors from the chick retina in culture. Here, by using an enriched retinal glial cell culture, we characterized some mechanisms underlying glial death promoted by cannabinoids. Methods and Results: Retinal cultures obtained from 8-day-old (E8) chick embryos and maintained for 12–15 days (C12–15) were used. MTT assays revealed that the CB1/CB2 agonist WIN 55,212-2 (WIN) decreased cell viability in the cultures in a time-dependent manner, with a concomitant increase in extracellular LDH activity, suggesting membrane integrity loss. Cell death was also dose-dependently induced by cannabidiol (CBD), Δ^9^-tetrahydrocannabinol (THC), and CP55940, another CB1/CB2 agonist. In contrast to WIN-induced cell death that was not blocked by either antagonist, the deleterious effect of CBD was blocked by the CB2 receptor antagonist SR144528, but not by PF514273, a CB1 receptor antagonist. WIN-treated cultures showed glial cells with large vacuoles in cytoplasm that were absent in cultures incubated with WIN plus 4-phenyl-butyrate (PBA), a chemical chaperone. Since cannabinoids induced the phosphorylation of eukaryotic initiation factor 2-alfa (eIF2α), these results suggest a process of endoplasmic reticulum (ER) swelling and stress. Incubation of the cultures with WIN for 4 h induced a ~five-fold increase in the number of cells labeled with the ROS indicator CM-H2DCFDA. WIN induced the phosphorylation of JNK but not of p38 in the cultures, and also induced an increase in the number of glial cells expressing cleaved-caspase 3 (c-CASP3). The decrease in cell viability and the expression of c-CASP3 was blocked by salubrinal, an inhibitor of eIF2α dephosphorylation. Conclusions: These data suggest that cannabinoids induce the apoptosis of glial cells in culture by promoting ROS production, ER stress, JNK phosphorylation, and caspase-3 processing. The graphical abstract was created at Biorender.com.

## 1. Introduction

The retina contains a key type of glia, the Müller cell, that spans the full thickness of the tissue, with its cell body located in the inner nuclear layer and its lateral processes expanding in the synaptic plexiform layers. In the mature tissue, Müller cells are involved in the control of extracellular milieu by, for example, regulating levels of K^+^, H^+^, and neurotransmitters. While this cell type expresses a variety of channels and transporters, they release several mediators known as gliotransmitters such as ATP, D-serine, and glutamate. Additionally, they express multiple neurotransmitter receptors, including those for nucleotides, glutamate, dopamine, and GABA, among others [1,2,3,4].

Müller glia cells are the last cell type generated in the retina and are recognized as a potential source of progenitors. Following cytotoxic damage that leads to the loss of a substantial number of retinal neurons, Müller cells can dedifferentiate and generate proliferating progenitor-like cells [5].

The two major endogenous ligands *N*-arachidonoylethanolamide (anandamide, AEA) and 2-arachidonoylglycerol (2-AG), along with their receptors and enzymes responsible for their synthesis and breakdown, constitute the functional endocannabinoid system present in neural tissue. In brief, while phospholipase D converts N-arachidonoyl phosphatidylethanolamine (NAPE) in AEA + phosphatidic acid, 2-AG synthesis occurs through the hydrolysis of diacylglycerol by DAG lipases (DAGL). AEA and 2AG are metabolized by several enzymes, especially fatty acid amide hydrolase (FAAH) and monoacylglycerol lipase (MGL) [6].

Cannabinoids can interact with at least two G-protein-coupled receptors—CB1 and CB2, both of which are negatively coupled to adenylate cyclase [7]. The CB1 receptor is the better-characterized subtype in the CNS. In neurons, its activation leads to adenylate cyclase inhibition, reduced cyclic AMP levels, PKA inhibition, blockade of voltage-gated calcium channels (VGCCs), and stimulation of rectifying K^+^ channels. This results in hyperpolarization of presynaptic terminals and reduced neurotransmitter release [8,9]. Conversely, in astrocytes, cannabinoid receptors can also couple to G_q_ protein. Their activation causes intracellular calcium mobilization and the release of gliotransmitters, such as glutamate [9,10]. Furthermore, in astrocytes and other cell types, CB1 receptors can activate mitogen-activated protein kinase (MAPK), including extracellular signal-regulated kinase (ERK), c-Jun-N-terminal kinase (JNK), and p38 kinase. CB1 receptors can also activate PI3K/Akt signaling pathway, which is responsible for CB1-induced protective effects on cultured astrocytes against ceramide-induced apoptosis [11,12].

The presence of cannabinoid ligands, receptors, and enzymes was demonstrated in the retina across various species, including the chick retina. Enzymes involved in cannabinoid degradation (FAAH, MGL, and COX-2) and synthesis (phospholipase D and DAGL) are found in almost all retinal neurons and glia [13]. Similarly, both CB1 and CB2 receptors are expressed in several types of neurons of the vertebrate retina, as well as in glia Müller cells [13,14,15,16]. Expression of both receptors was also identified in the chick embryo retina [17] and developing retinal neuronal and glial progenitors in culture [18,19]. In chick retinas, cannabinoids induce the dedifferentiation of Muller cells and proliferation of glia-derived progenitors [20]. Moreover, in chick embryo retinal cultures, the cannabinoid agonist WIN 55,212-2 (WIN) reduces cyclic AMP formation and GABA release from retinal cells in culture [18].

Cannabinoids exert a modulatory role in synaptic transmission by reducing the release of neurotransmitters such as glutamate from presynaptic terminals. This negative modulatory effect was proposed as the underlying mechanism of the CB1 receptor-mediated neuroprotection against excitotoxicity in the brain [21]. Neuroprotective effects of cannabinoids against excitotoxicity were also described in the retina. For example, AMPA-induced excitotoxicity to horizontal cells and bNOS, ChAT, and GABA-expressing amacrine cells is mitigated when the endocannabinoid anandamide, the CB1/CB2 agonist HU-210, and the CB1 agonist R1-methanandamide (MethAEA) are intravitreally co-administered with the glutamatergic agonist [22]. Moreover, both tetrahydroxycannabinol (THC) and cannabidiol (CBD), as well as the synthetic cannabinoid WIN 55,212-2, protect the retina against NMDA-induced excitotoxicity [23,24]. Since no effect of a CB2 receptor agonist was observed against AMPA-induced excitotoxicity in this tissue, the CB1 is postulated to be the receptor responsible for the neuroprotective effect of cannabinoids against AMPA excitotoxicity in the retina. In this excitotoxicity model, 2AG also protects retinal neurons via the CB1 receptor and PI3K/Akt signaling pathway [22].

Neuroprotection by CB1 receptor activation was also highlighted in retinal disease models. In an in vivo model of glaucoma, intravitreal injection of the anandamide stable analog R(+)-methanandamide (MethAEA) rescues retinal ganglion cells (RGCs) via activation of CB1 receptors and TRPV1 channels [25]. In a streptozotocin model of diabetic retinopathy, CBD treatment provides neuroprotection to inner retinal neurons [26]. Activation of the CB1 receptor by the agonist HU210 protects cholinergic amacrine and rod bipolar cells during chemical ischemia [27].

As opposed to the neuroprotective role of cannabinoids mentioned above, some previous data pointed to a cytotoxic effect of these compounds. Treatment of hippocampal neurons in culture with THC induces a time- and dose-dependent decrease in cell viability, possibly mediated by phospholipase A_2_–cyclooxigenate pathway [28]. Systemic administration of THC retinal damage occurs by inducing inflammation, oxidative stress, and increased apoptosis of photoreceptors [29].

In some retinal disease models, the cytotoxic effect of cannabinoids was also detected. Treatment of streptozotocin-induced diabetic mice with the CB1 receptor antagonist SR141716A (rimonabant) or genetic ablation of CB1 receptors prevents the death of retinal vascular cells in the retina by attenuating oxidative stress and the activation of the pro-apoptotic p38/Jun N-terminal kinase [30]. In retinal pigmented epithelium cells, CB2 receptor activation increases the production of IL-6 and IL8 pro-inflammatory cytokines and photoreceptor degeneration [31]. CB1 receptor antagonist SR141716A also prevents photoreceptor cell death in a light-induced retinal degeneration model [32].

Cannabinoids can also exert deleterious effects in neurons and astrocytes from the developing CNS. In neonatal rat cortical neurons and astrocytes, CBD induces apoptosis in a dose- and time-dependent manner [33]. While the synthetic WIN 55,212-2 decreases cell proliferation in chick embryo retina cell cultures, it induces the death of 2M6-positive cells, indicating that cannabinoids are cytotoxic to retinal glial progenitors [19]. In the present work, we used an enriched chick retina glial cell monolayer culture to characterize the mechanisms underlying the harmful effect of synthetic cannabinoids and the two phytocannabinoids CBD and THC on glial Müller cells. Our findings reveal that cannabinoids increase intracellular levels of ROS and endoplasmic reticulum (ER) swelling, promote phosphorylation of eIF2α and JNK, activate caspase-3, and induce retinal glia death.

## 2. Materials and Methods

### 2.1. Materials

Fertilized eggs from white Leghorn chickens (*Gallus gallus domesticus*) were purchased from a local supplier. Monoclonal mouse anti-2M6 antisera specific for avian retinal Müller glial cells [34] was generously provided by Dr. B. Schlosshauer (Max-Planck-Institute, Tübingen, Germany). WIN 55,212-2 (WIN), Salubrinal, and mouse anti-α-tubulin were purchased from Sigma-Aldrich (São Paulo, Brazil); 4-phenyl-butyrate (4-PBA), cannabidiol (CBD), Δ^9^-tetrahydrocannabinol (THC), and CP 55,940 were from Cayman Chemical Co. (Ann Arbor, MI, USA) Rabbit anti-cleaved-caspase 3 (catalog # 9664), anti-alfa-tubulin (catalog # 3873), and anti-phospho-SAPK/JNK (catalog. # 4668) were from Cell Signaling Technology (Danvers, MA, USA). Anti-phospho-P38 (catalog # sc-17852) was from St. Cruz Biotech (Dallas, TX, USA). MEM, fetal bovine serum, CM-H2DCFDA, Alexa Fluor 488 goat anti-mouse, and Alexa Fluor 568 goat anti-rabbit were from Thermo Fisher Scientific (São Paulo, Brazil). All other chemicals used were of analytical grade.

### 2.2. Retinal Glial Cell-Enriched Monolayer Cultures

Embryos were staged as described by [35] and retinal cell monolayer cultures prepared following the protocol in [36] with minor adjustments. Briefly, 8-day-old embryos (E8) were euthanized by decapitation and the eyes transferred to Ca^2+^- and Mg^2+^-free balanced salt solution (CMF) for retina dissection. Tissues were treated with trypsin (0.1%) at 37 °C for 20–25 min. After trypsin removal, tissues were resuspended in minimum essential medium (MEM) supplemented with 5% fetal calf serum, 2 mM glutamine, 100 U/mL penicillin, and 100 mg/mL streptomycin, then mechanically dissociated and the cells counted. To enrich for glial cells, dissociated cells were seeded and cultivated at 37 °C in a humidified atmosphere of 95% air/5% CO_2_ for 10–14 days (E8C10–E8C14). During this time, cultured neurons progressively aggregated (Appendix A) and died, while glial progenitors adhered, proliferated, and became flat by expanding their cytoplasm. After 13–14 days, most of the cultured cells were positive for the glial-specific antigen 2M6, but negative for the neuronal marker beta-tubulin-III. For 3-(4, 5-dimethylthiazolyl-2)-2, 5-diphenyltetrazolium bromide (MTT) or extracellular lactate dehydrogenase (LDH) activity experiments, cells were seeded on 24-well dishes at a density of 1.5 × 10^6^ cells/well. For immunofluorescence, cells were seeded on coverslips inside culture dishes at a density of 5 × 10^6^ cells/dish (5.2 × 10^3^ cells/mm^2^). For Western blotting experiments, 10^7^ cells were seeded on 35 mm culture dishes (1.04 × 10^4^ cells/mm^2^). Culture medium was changed every other day. Retinal cultures were treated with cannabinoids for 3 h (H2DCFDA for detection of ROS) or 24 h (viability, Western blotting, and immunofluorescence experiments).

### 2.3. MTT Viability Assay

The viability of cells was evaluated using the MTT reduction method [37], adapted for chick retinal cultures by [38]. Briefly, cells seeded in 24-well dishes (1.5 × 10^6^ cells/well) were incubated at 37 °C with MTT (1 mg/mL) in MEM buffered with 25 mM HEPES, pH 7.4. After 20 min, medium was replaced with a mixture of HCl/propanol to dissolve formazan crystals. Absorbance was determined at 550 nm after subtracting the absorbance at 650 nm. Four wells of the multi-well dish were used for each treatment. Controls were always determined in the dish and received the maximal concentration of vehicle used to dissolve the drugs that were used in the experiment. The concentrations of vehicle were adjusted and equal in all wells of the dish. Results were expressed as arbitrary units (a.u.) of absorbance.

### 2.4. Extracellular LDH Activity Assay

Extracellular LDH activity was determined by using the LDH Liquiform Kit according to the manufacturer’s instructions (Labtest, Lagoa Santa, MG, Brazil). Briefly, samples of the culture medium (15 µL) were incubated in black 96-well dishes with 200 µL of a mixture of NADH and pyruvate. Oxydation of NADH in samples was measured by UV absorbance (340 nm) in a BioTek Synergy H1 plate reader (Agilent Tech., Barueri, SP, Brazil). Absorbance was measured at 1 min (A1), 3 min (A2), and 5 min (A3) after the addition of the reagents in order to verify linearity of the reaction. The decreases in absorbance (A1 − A2)/2 or (A1 − A3)/4 were calculated. LDH activity was expressed as arbitrary units/min/culture. Culture medium containing 5% serum was used as control (blank).

### 2.5. Immunocytochemistry

Coverslips with retinal cultures from E8 embryos maintained for 12–14 days (E8C12–14) were rinsed with phosphate-buffered saline (PBS) and fixed in 4% paraformaldehyde in 0.16 M phosphate buffer (pH 7.6) for 15 min. Following three 5 min washes with PBS, cells were permeabilized with 0.3% triton X-100 for 30 min. Nonspecific labeling was inhibited by incubating cells for 60 min in PBS with 0.3% Triton X-100 plus 5% NGS. Cells were incubated overnight at 4 °C with anti-2M6 (1:200), anti-cleaved caspase-3 (Asp175) (1:100), or anti-alfa-tubulin (1:2000) primary antibodies diluted in PBS with 0.3% Triton X-100 and 1% BSA. Primary antisera were removed, and the coverslips were washed and incubated with Alexa secondary antibody (1:200) for 2 h at room temperature. Nuclei were labeled with DAPI or Hoechst 34580 (Thermo Fisher Brazil, São Paulo, SP, Brazil), and the coverslips were observed and imaged with a Leica SP5 confocal microscope.

### 2.6. Western Blotting

Protein extracts from control or treated retinal cultures were prepared by lysing cells in sample buffer (62.5 mM Tris-HCl, pH 6.8, 10% glycerol, 2% sodium dodecyl sulfate (SDS), and 5% 2-mercaptoethanol) without bromophenol blue. After boiling for 10 min, protein content in 2 μL samples was determined by the Bradford assay (1976), using BSA solution plus 2 μL sample buffer as standard. Protein extracts with 0.02% bromophenol blue (50 µg/lane) were resolved on 9% SDS-PAGE and transferred to PVDF membranes (Thermo Fisher Brazil, São Paulo, SP, Brazil). Membranes were stained with Ponceau red and blocked in Tris-buffered saline, pH 7.6 (TBS), with 0.1% Tween-20 (TTBS) and 5% non-fat milk. Membranes were incubated with anti-phospho-SAPK/JNK (1:1000) or anti-phospho-P38 (1:1000) overnight at 4 °C. Membranes were washed 3 × 5 min with TTBS and incubated with HRP–conjugated secondary antibody (1:4000 in TTBS; Sigma-Aldrich) for 2 h at room temperature. Following 2 washes with TTBS and 1 wash with TBS, labeling was detected via ECL, according to the manufacturer’s protocol (ECL prime, Thermo Fisher). Membranes were stripped with 0.2 M glycine, pH 2.2, washed 3 × 10 min with TBS and re-probed with anti-α-tubulin antiserum (1:100,000 in TTBS; Sigma-Aldrich) for 1 h at room temperature, rinsed in TTBS, and incubated with anti-mouse HRP–conjugated secondary antibody for 45 min at room temperature. Following three washes (10 min each), bands were detected with ECL. Band intensities were quantified using TotalLab TL120 1D software.

### 2.7. Detection of ROS by CM-H2DCFDA Fluorescence

Production of intracellular reactive oxygen species (ROS) was assessed in live cells using the CM-H2DCFDA (Thermo Fisher) probe. Cultures (E8C13) were treated with cannabinoids for 3 h, washed 3 times with Hanks’ balanced salt solution (128 mM NaCl, 4 mM KCl, 1 mM Na_2_HPO_4_, 0.5 mM KH_2_PO_4_, 1 mM MgCl_2_, 3 mM CaCl_2_, 20 mM HEPES, 12 mM glucose, pH 7.4), and incubated with 10 μM CM-H2DCFDA in Hanks’ solution for 20 min at 37 °C in the dark. In the last 5 min of incubation, cell nuclei were labeled with Hoechst 34580. Cells were washed three times with Hanks’ solution and coverslips were mounted in an imaging chamber (Warner Instruments, Hamden, CT, USA). Glia fluorescent cells were observed and photographed with a confocal SP5 microscope using the 488 laser line and an emission bandwidth of 503–558 nm for H2DCFDA fluorescence. The 405 laser line was used to determine Hoechst-positive nuclei. At least 10 fields were photographed for each coverslip. The proportion of ROS-positive cells was obtained by dividing the number of green fluorescent cells/Hoechst nuclei in each field.

### 2.8. Data Analysis

Results were analyzed using ANOVA and Bonferroni’s multiple-comparisons test with GraphPad Prism 8. Significance was set at *p* < 0.05, and the results are expressed as the mean ± S.E.M of 3 or more independent experiments conducted in duplicate or triplicate.

## 3. Results

Recently, we have shown that the synthetic cannabinoid WIN 55,212-2 is able to induce, in a dose-dependent manner, the death of glial progenitors in developing retinal cultures at E7C2 [19]. In order to verify whether cannabinoids could also affect the viability of more differentiated Müller glia cells in culture, the effect of WIN on cell viability in cultures at E8C13 was investigated. The effect of the phytocannabinoids CBD and THC, as well as the synthetic compound CP 55940 was also studied (Figure 1). Cultures were incubated with 1 µM WIN or increasing concentrations of CBD (Figure 1A), THC (Figure 1B), or CP 55940 (Figure 1C) for 24 h before MTT assay. A dose-dependent loss of cell viability was observed with the three compounds, and significant decreases were observed using concentrations of 10 µM or higher of the cannabinoids.

The decrease in cell viability induced by 1 µM WIN (Figure 1D) or 10 µM CBD (Figure 1E) was also time dependent. Significant decreases in cell viability were observed after 20 h or 24 h of incubation with WIN (~24.4% and 35%) or CBD (~73% and 84%). The decrease in MTT reduction observed in cultures incubated with 1 µM WIN and 10 µM CBD was accompanied by an increase in LDH activity in culture medium. After incubations with WIN or CBD for 24 h, LDH activity increased approximately 6- and 9.5-fold in the medium, respectively.

To determine whether the decrease in cell viability induced by cannabinoids was mediated by the activation of CB1 or CB2 receptors, the effects of the CB1 receptor antagonist PF514273 and the CB2 receptor antagonist SR144528 were investigated (Figure 2). A concentration of 30 µM of the antagonists was used. No attenuation in WIN-induced decrease in cell viability was observed with either antagonist. In contrast, a significant reduction in the CBD-dependent decrease in cell viability was observed with the CB2 receptor antagonist SR144528. Furthermore, although not statistically significant, a tendency toward a further decrease in the effect of CBD on cell viability was observed with the CB1 receptor antagonist. Similar results were obtained with a concentration of 20 µM of the antagonists.

WIN-treated cultures at E8C13 showed the majority of glial cells with large vacuoles in their cytoplasm (Figure 3) that could not be labeled with 5 µM quinacrine, discarding that they were acidic vesicles or lysosomes. CBD- and THC-treated cultures also presented cytoplasmic vesicles. However, both the number and the size of these vacuoles were smaller than those observed in WIN-treated cultures.

Cannabinoids can induce stress in the endoplasmic reticulum (ER) followed by the unfolded protein response (UPR) in tumor cells, such as breast and non-melanoma skin cancer cells [39,40], as well as endometrial stromal and choriocarcinoma cells [41,42], among others. In order to characterize whether the cytoplasmic vacuoles observed in WIN-treated glial cells were due to ER stress induced by the cannabinoid, coverslips with cultures at E8C12 were treated with 1 µM WIN in the presence or absence of 50 µM 4-phenyl-butyrate (4-PBA), a chemical chaperone that interacts with hydrophobic segments of unfolded proteins, preventing protein aggregation, promoting protein folding, and inhibiting the expression of proteins induced by UPR. After 24 h, cultures were fixed and labeled for 2M6, an avian glial-specific marker (Figure 4A). As can be noticed, glial cells from cultures treated with 1 µM WIN showed large vesicles (arrow) mainly at the periphery of the cytoplasm and with the intracellular 2M6 labeling accumulated near the cell nucleus. No visible structures inside vacuoles were noticed. In contrast, treatment of the cultures with WIN in the presence of 50 µM 4-PBA substantially decreased the number and size of vacuoles inside glial cells. Also, 2M6 labeling was much more uniform in glial cytoplasm. Cultures treated only with 4-PBA showed 2M6 labeling similar to the control cultures.

ER homeostasis can be disrupted by several conditions, such as nutrient depletion, viral infections, fluctuations in calcium, inflammatory cytokines, and oxidative stress, among others [43,44]. Since some cannabinoids such as CBD induce reactive oxygen species [45], the effect of WIN, CBD, and THC on intracellular ROS content was investigated. Cells cultivated on coverslips (E8C10–12) were incubated with 1 µM WIN or 10 µM CBD or 10 µM THC for 3 h, washed 3× with 1 mL Hanks’ solution, and incubated with 10 µM of the CM-H2DCFDA probe for 20 min at 37 °C. Cells were quickly photographed for H2DCFDA and Hoechst fluorescence on a confocal microscope and counted. WIN induced a significant increase in the percentage of ROS-positive glial cells (Figure 4B,C). While 11.3 ± 1.2% of glial cells were labeled for H2DCFDA in control cultures, 60.8 ± 6.1% of glial cells were labeled in the WIN-treated cultures. Increases of 25.1 ± 5.6% and 21.8 ± 5.6% in the percentage of H2DCFDA-labeled cells were also observed in CBD- and THC-treated cultures, respectively. However, these increases were not statistically significant.

UPR involves the activation of three main ER resident membrane proteins, including protein kinase RNA-like ER kinase (PERK) that phosphorylates the eukaryotic initiation factor 2-alfa (eIF2α), causing inhibition of the general cap-dependent protein translation [46]. In order to verify whether cannabinoids induced UPR in the cells, retinal cultures at E8C13 were treated with WIN, CBD, or THC for 24 h, and the amount of phospho-eIF2α was determined by Western blotting (Figure 5A,B). While WIN induced a ~2.3-fold stimulation, CBD and THC also significantly increased eIF2α phosphorylation by 2.1- and 1.8-fold, respectively. To determine whether CBD could induce ER stress and the phosphorylation of eIF2α at a lower concentration, the effect of 3 µM CBD alongside 10 µM CBD on p-eIF2α levels was investigated (Figure 5C–E). While a 10 µM concentration of CBD induced ~2-fold stimulation, 3 µM CBD promoted an almost 1.4-fold increase in p-eIF2α levels.

Under UPR, ER resident protein IRE1α dimerizes and auto-phosphorylates, leading to several events, such as the activation of c-Jun N terminal kinase (JNK) [47]. In order to verify whether the detected ER stress induced by cannabinoids activated JNK, cultures at E8C13 were incubated with 1 µM WIN or 10 µM CBD and 10 µM THC for 24 h, and the amount of phospho-JNK and phospho-p38 (another stress-activated kinase) was determined by Western blotting. As shown in Figure 6, an increase of almost five-fold in the phosphorylation of JNK, but not in p-38 phosphorylation, was observed in WIN-treated cultures. Both CBD and THC also increased JNK phosphorylation (Figure 6D,E). No effect of CBD or THC on the levels of phosphorylated p-38 was detected.

ER stress and JNK activation can influence cell death machinery to process caspase and execute apoptosis [46]. In order to verify whether cannabinoid-dependent ER stress induced glial cell apoptosis, cultures at E8C13 were treated with 1 µM WIN for 24 h, and then were fixed and processed for detection of cleaved-caspase 3 (c-CASP3) by immunocytochemistry (Figure 7). While in control cultures, c-CASP3-labeled cells were rarely noticed, while ER-stressed glial cells presenting c-CASP3 labeling were quite frequently noticed in WIN-treated cultures.

GADD34 is a protein that interacts with protein phosphatase 1 (PPL1) that dephosphorylates eIF2α and causes the release of the translational block. GADD34 expression correlates with apoptosis by yet-unknown mechanisms, and inhibition of eIF2α dephosphorylation by salubrinal reduces ER stress-mediated apoptosis [48]. The effect of salubrinal on the decrease in cell viability induced by WIN is shown in Figure 8. Cultures at E8C13 were incubated with 1 µM WIN in the presence or not of 50 µM salubrinal for 24 h. Both the decrease in cell viability (Figure 8A) and the increase in extracellular LDH activity (Figure 8B) were significantly attenuated by salubrinal. Moreover, the expression of c-CASP3 in 2M6^+^ glial cells observed in cultures at E8C14 that were treated with WIN for 24 h was also attenuated (Figure 8C). Very few glial cells labeled for c-CASP3 were detected in the cultures treated with 1 µM WIN plus 50 µM salubrinal.

## 4. Discussion

Previously, in developing retinal cell cultures, we detected a decrease in glia progenitor viability promoted by the cannabinoid agonist WIN 55,212-2 [19]. In the present work, we characterized the effect of synthetic and phytocannabinoids on cell viability in a glia-enriched retinal cell culture model. Our data show that cannabinoids in a concentration-dependent manner significantly reduce the viability of cultured cells. Moreover, the cytotoxic effects of WIN and CBD are time dependent, with an increase in LDH activity in the culture medium 6–8 h after the addition of the cannabinoid, suggesting that the decrease in cell viability in the cultures is accompanied by late loss of cellular plasma membrane integrity. Accordingly, previous work showed that the sub-micromolar concentrations of CBD decrease the viability of cultured cortical astrocytes and neurons [33], as well as the viability of glioblastoma cells [49]. Interestingly, a recent study revealed a nuanced effect: while micromolar concentrations of CBD were toxic to both mouse neurons and astrocytes in culture, nanomolar concentrations offered protection against hydrogen peroxide (H_2_O_2_)-induced cell death specifically in neurons, but not astrocytes. This neuroprotective effect of CBD was observed only against oxidative stress, but not against glutamate-mediated excitotoxicity, highlighting that CBD’s actions are cell type- and insult-specific [50].

The pharmacology of cannabinoids is complex, with CB1 and CB2 receptors as well as various receptors such as GPR55 and TRPV1 mediating different cannabinoid actions [51]. CBD is considered a negative allosteric modulator (NAM) at CB1 receptors, a partial agonist at CB2 receptors, and an agonist at TRPV1 channels [51,52]. The present work reveals that the effect of CBD is significantly attenuated by the co-incubation of the cells with the CB2 receptor antagonist SR144528, but not by the co-incubation with PF 514273, a CB1 receptor antagonist. This result suggests that the decrease in cell viability induced by CBD is mediated by the activation of CB2 receptors in cultured glial cells. Moreover, this result also suggests that the deleterious effect of CBD is not due to drug toxicity from the high concentration of this cannabinoid used in the present study. However, no effect of SR144528 on the WIN-mediated decrease in cell viability was detected, suggesting that other receptors may participate in the cannabinoid-dependent decrease in glial cell viability. Here, we performed experiments with capsazepine and CID16020046, two antagonists for TRPV and GPR55 receptors, respectively, but no attenuation of WIN- or CBD-mediated decrease in cell viability was observed. Future experiments using combinations of different antagonists or agonists could help further clarify the receptors involved in the deleterious effect of cannabinoids on cultured glial cells.

The endoplasmic reticulum is an organelle rich in chaperones that bind to newly synthetized proteins, ensuring their proper folding and/or maturation while preventing aggregation [53]. Abnormal accumulation of misfolded proteins in the ER lumen triggers ER stress followed by the activation of UPR to restore homeostasis. Unresolved UPR activation in astrocytes is associated with inflammatory gene expression and reduced trophic support to neurons, and is potentially related to the non-resolving nature of certain neurological diseases [54]. ER stress, induced by factors such as nutrient deprivation, viral infections, and disruption in cellular Ca^2+^ and/or ROS homeostasis, can result in ER expansion and the formation of cytoplasmic vacuoles due to driving osmotic pressure pulling water from the cytoplasm [55]. The present results reveal that cultures treated with WIN for 24 h display glial cells containing large vesicles or vacuoles within their cytoplasm, whereas CBD- or THC-treated cultures show only small vacuoles. The compound 4-PBA, which is a well-known chemical chaperone [56], substantially decreases both the number and the size of vacuoles inside glial cells in cultures exposed to WIN, strongly suggesting that this cannabinoid induces swelling of the glial ER. This hypothesis is further supported by the observation that cannabinoids induce the phosphorylation of eIF2α, a protein phosphorylated by PERK after ER stress. Similar to our results are the observations that cannabinoids including WIN 55,212-2 induce ER dilation and cytoplasm vacuolization in tumor cells [57]. CBD-dependent cytoplasm swelling was also detected in primary cultures of rat astrocytes [33]. However, the differences in vacuole size observed in cultures treated with WIN, CBD, and THC still need clarification. It is well known that complex regulatory crosstalk between UPR and autophagy occurs after ER stress. Thus, it is possible that the different cannabinoids used here regulated UPR/autophagy interaction with distinct efficiency or specificity. The use of specific antibodies or probes to label the different cellular compartments could help to clarify this point.

ER stress can be induced by different conditions, including elevated ROS production [43,44]. Our data show that treating retinal glia-enriched cultures with cannabinoids induces an increase in the number of glial cells expressing high levels of ROS-sensitive dye fluorescence. Moreover, the percentage of cells with increased levels of ROS is higher in WIN-treated cultures than those stimulated with CBD or THC, suggesting that the oxidative stress induced by WIN is higher or faster than the one promoted by CBD or THC. Based on these observations, it seems reasonable to suppose that at least the WIN-mediated decrease in glia viability is related to the increase in oxidative and ER stress in the cultured glia cells. However, the underlying molecular mechanisms activated by cannabinoids that generate these stress responses deserve to be further explored. An interplay between ROS and ER stress was detected in cannabinoid-induced cell apoptosis in different cell types, including placental trophoblasts [42] and tumor cells [43].

The increase in the number of glial cells expressing high levels of ROS-sensitive dye fluorescence was observed in cultures incubated with cannabinoids for 3 h. Previously, Kubrusly and co-workers [18], by using chick retinal cell cultures at E8C8 that still contained both neurons and glial cells, showed that a 3 h incubation of the cultures with WIN significantly decreased (~80%) the number of cells responding to ATP, with increases in intracellular Ca^2+^ levels. Since ATP-induced intracellular calcium increase occurs only in glial cells, and not in neurons that are not affected by WIN, these data all together suggest that cannabinoid-mediated increases in oxidative stress are concomitant with the dysregulation of Ca^2+^ homeostasis in cultured glial cells. The use of specific ROS scavengers or intracellular calcium chelators could help clarify the relationship between calcium homeostasis dysregulation and ROS formation in the cannabinoid-treated glial cells.

Three distinct pathways to apoptosis are known: receptor-mediated pathway, mitochondrial pathway, and ER stress induced pathway [58]. Our present results show that WIN promotes the phosphorylation of JNK, but not of P38. Both CBD and THC also induce the phosphorylation of JNK, suggesting that cannabinoids activate this MAP kinase in the retinal glial cell cultures. Our data also show that WIN induces the expression of cleaved CASP-3 in glial cells in the cultures, suggesting that cannabinoids decrease the viability of cultured glia cells by inducing apoptosis. Since our data show that cannabinoids induce ER stress in cultured glial cells, it is reasonable to propose that cannabinoids activate ER stress-induced pathway of apoptosis, which may involve the activation of JNK pathway. The UPR that follows ER stress involves the activation of IRE1α [44], which dimerizes and auto-phosphorylates, leading to events such as the activation of tumor necrosis factor receptor-associated factor 2 (TRAF2), which can stimulate apoptosis signal-regulating kinase 1 (ASK1)/JNK pathway [47]. Accordingly, previous data showed that the activation of JNK is involved in the apoptosis of cultured astrocytes induced, for example, by ethanol and ceramide [59,60]. It can also activate pro-apoptotic proteins BAX and BAK and inhibit the protective protein Bcl-2 in tumor cells [61,62,63]. In the retina, activation of JNK is involved in retinal cell excitotoxicity, and its inhibition attenuates cell death in this tissue [64,65,66,67]. Furthermore, cannabinoids induce cell death associated with JNK phosphorylation in certain cell types, including PC12 cells and colorectal cancer cells [68,69]. Here, we performed experiments with pharmacological inhibitors of JNK activity to prevent cannabinoid-dependent decrease in cell viability, but no consistent attenuation in the decreased cell viability was obtained. Thus, the role of JNK in cannabinoid-induced decrease in glial viability is a point that requires clarification. RNAi and similar techniques could be helpful to define this point.

Phosphorylation of eIF2α promotes the blockade of cap-dependent translation in cells under ER stress. It has a major role in the cytoprotective gene expression program known as the integrated stress response (ISR), which is responsible for the ability of cells to adapt to ER stress [70,71,72,73]. The interaction of GADD34 with protein phosphatase 1 (PPL1) dephosphorylates p-eIF2α, causing the release of the translational block and cell death. Salubrinal is a small molecule that inhibits p-eIF2α dephosphorylation and protects cells against ER-stress mediated apoptosis [48]. The present results show that salubrinal is capable of blocking the decrease in cell viability as well as the increase in cleaved CASP3 expression in cultured glial cells exposed to WIN 55,212-2, suggesting that phosphorylation of eIF2α increases the survival of glial cells under cannabinoid-dependent ER stress. Accordingly, previous studies have shown that salubrinal offers neuroprotection to the cortex and hippocampus by preventing ER stress-induced apoptosis caused by traumatic brain injury [74,75] or kainate-mediated excitotoxicity [76]. In the retina, salubrinal enhances photoreceptor survival in the P23H-1 transgenic rat model of retinitis pigmentosa, where mutated rhodopsin misfolding and aggregation lead to ER stress and UPR activation [77].

## 5. Conclusions

The data presented here show that cannabinoids have a deleterious effect on cultured retinal cells. They suggest that cannabinoids induce the apoptosis of glial cells in culture by promoting ROS production, ER stress, JNK phosphorylation, and caspase-3 processing. Moreover, our data support the idea that the therapeutical use of cannabinoid agonists needs to be implemented with caution, as they can produce undesirable, toxic effects on glial cells. It is well known that persistent activation of UPR after ER stress in astrocytes is related to direct CNS inflammation and has been implicated in several CNS diseases, including Alzheimer’s disease, multiple sclerosis, and Huntington’s disease, among others [54]. Thus, by characterizing the effects of cannabinoids in different cell populations and the underlying mechanisms, together with the development of more specific agonists and antagonists, will certainly contribute to the use of cannabinoids as therapeutic compounds.

## Figures and Tables

**Figure 1 brainsci-15-00291-f001:**
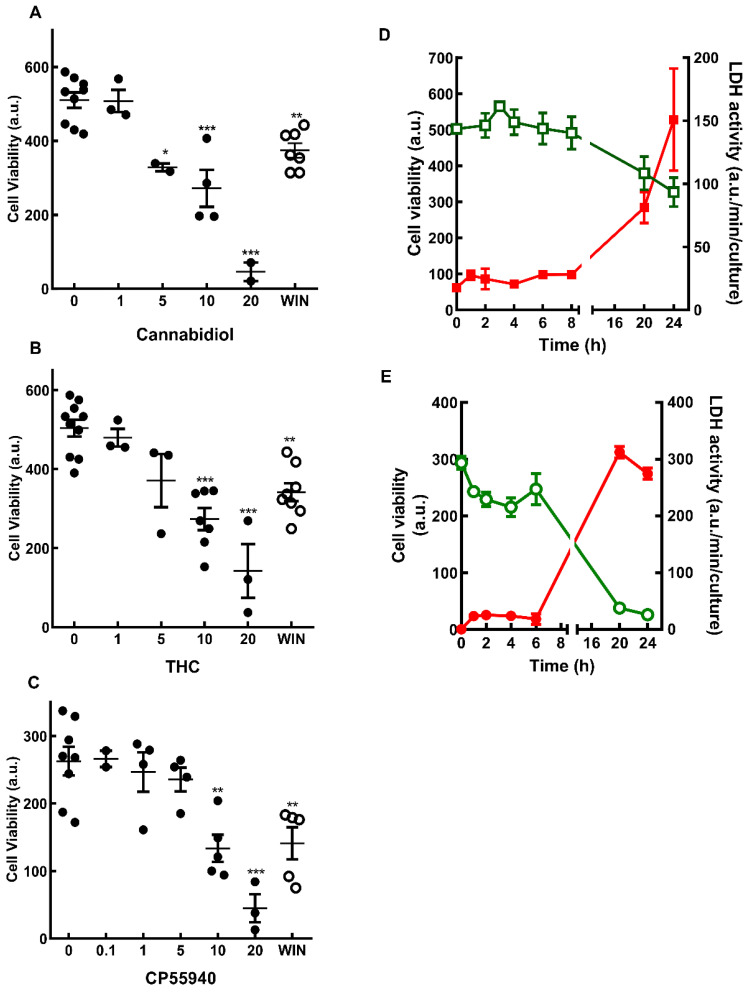
Cell viability in cultures treated with cannabinoids. Glia-enriched cultures at E8C13 were treated with µM concentrations of cannabidiol (**A**), THC (**B**), CP55940 (**C**), 1 µM WIN 55,212-2 (open circles) for 24 h or 1 µM WIN 55,212-2 (**D**), or 10 µM CBD (**E**) for increasing periods of time. Cultures were processed for MTT viability assays (**A**–**C**, and green open symbols and curves in **D**,**E**) or LDH activity (filled red symbols and curves in **D**,**E**) as described in the methods section. Data represent the mean ± S.E.M. of 2–10 experiments performed in triplicate (**A**–**C**) or 3 separate experiments performed in triplicate or quadruplicate (**D**,**E**). ANOVA and Bonferroni’s multiple-comparisons test were used. *** *p* < 0.001, ** *p* < 0.01, and * *p* < 0.05 as compared to control cultures (0 µM). a.u. = arbitrary units.

**Figure 2 brainsci-15-00291-f002:**
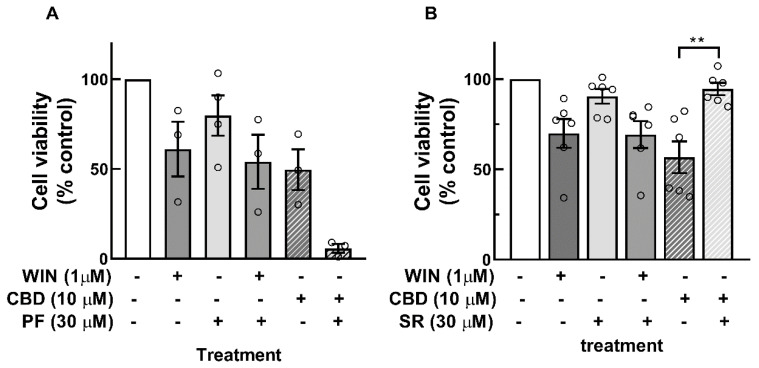
Effect of CB1 and CB2 receptor antagonists on the decrease in cell viability induced by WIN or CDB in glia-enriched cell cultures. Cultures at E8C13 were incubated with 1 µM WIN or 10 µM CBD in the presence or absence of 30 µM of the CB1 receptor antagonist PF514273 (**A**) or 30 µM of the CB2 receptor antagonist SR144528 (**B**) for 24 h. Cell viability was estimated by MTT assay. Data represent the mean % of control ± S.E.M. of 3 to 6 experiments performed in triplicate or quadruplicate. ANOVA and Bonferroni’s multiple-comparisons test was used. ** *p* < 0.01.

**Figure 3 brainsci-15-00291-f003:**
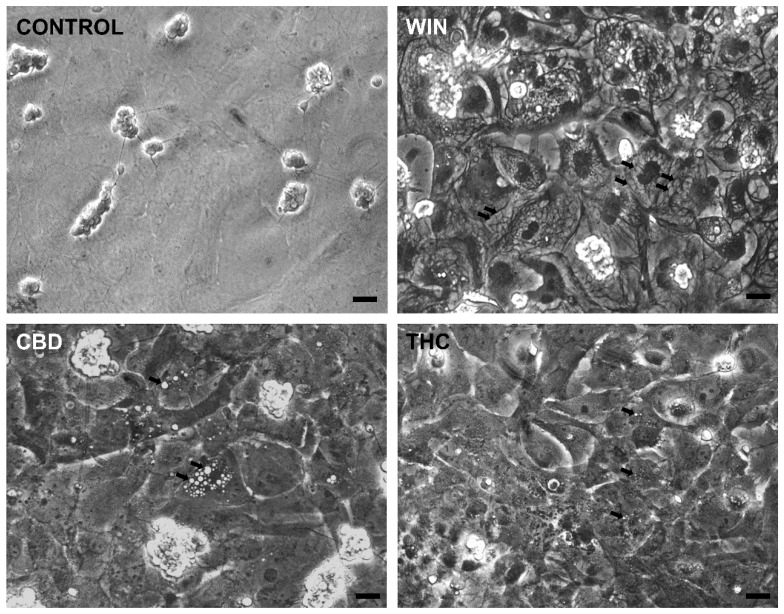
Cannabinoid agonists induce cytoplasmic vacuolization in retinal glial cells in culture. Glia-enriched cultures at E8C13 were treated with 1 µM WIN, 10 µM CBD, or 10 µM THC for 24 h, and then were fixed and imaged under phase contrast illumination. Note the big cytoplasmic vacuoles in WIN-treated cultures and smaller vacuoles in CBD- or THC-treated cultures (arrows). Bar = 20 µm.

**Figure 4 brainsci-15-00291-f004:**
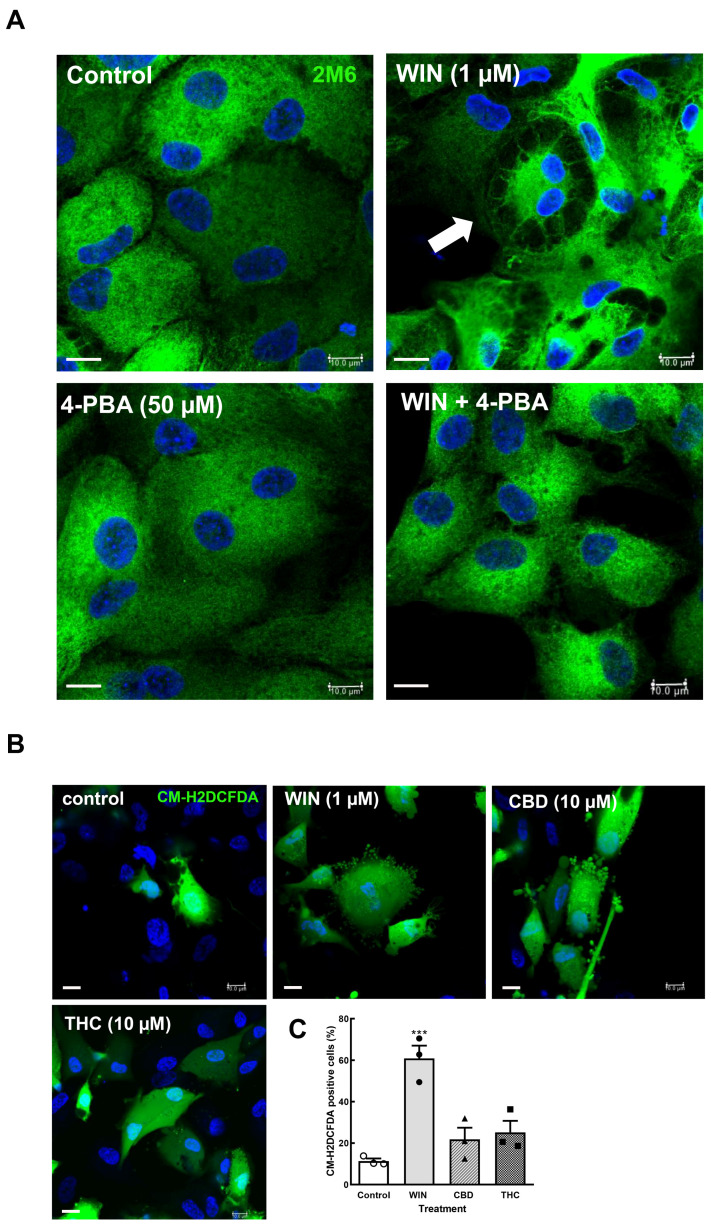
4-Phenyl-butyrate inhibits intracellular vacuolization in cultures treated with WIN 4-phenyl-butyrate (**A**). Glia-enriched cultures on coverslips at E8C13 were incubated with 1 µM WIN or with WIN plus 50 µM 4-PBA for 24 h, and then were fixed and processed by imunnocytochemistry for the avian glial-specific antigen 2M6. Hoechst was used to stain cell nuclei (blue). Cultures were visualized and photographed on a Leica SP5 confocal microscope. Note the large vacuoles in the WIN-treated cultures (arrows). Imunnofluorescence assays were conducted 3 times, with similar results. CM-H2DCFDA oxidation assays in retinal cultures (**B**,**C**). Glia-enriched cell cultures on coverslips at E8C10–E8C12 were treated with 1 µM WIN, 10 µM CBD, or 10 µM THC for 3 h, and then washed and incubated with the CM-H2DCFDA probe. Ten fields along the diameter of the coverslip were immediately photographed on a Leica SP5 microscope using a 40× objective and 405 and 488 laser lines following the manufacturer’s instructions. Hoechst was used to label cell nuclei (blue) and estimate the total number of cells in each field. The number of CM-H2DCFDA-labeled cells was divided by the number of cell nuclei in each field and expressed as the percentage of labeled cells (**C**). From 750 to 1030 nuclei were counted in each experiment, and data represent the mean ± S.E.M. of 3 separate experiments performed in duplicate. ANOVA and Bonferroni’s multiple-comparisons test were used. *** *p* < 0.001 versus control. Scale bar = 10 µm.

**Figure 5 brainsci-15-00291-f005:**
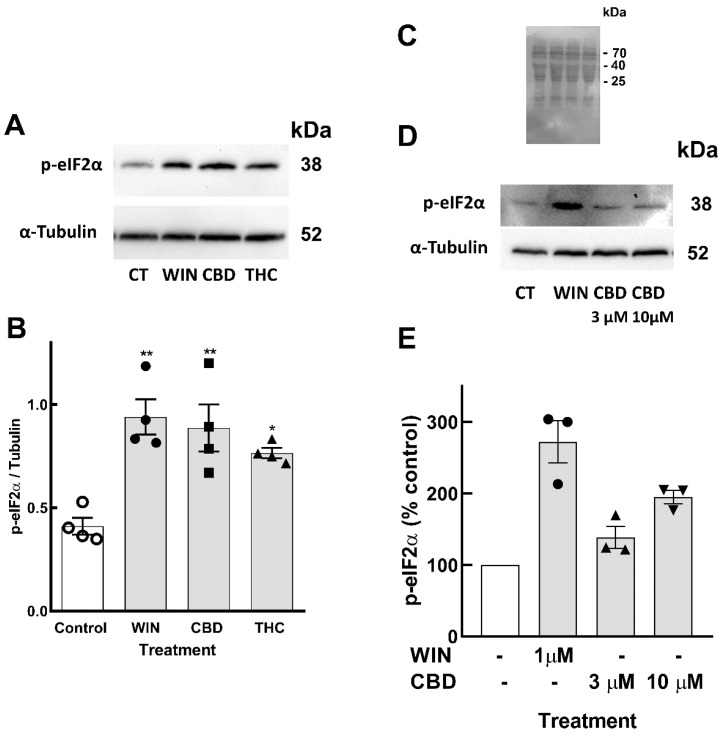
Phosphorylation of eIF2α induced by cannabinoids in retinal glia cultures. Glia-enriched cultures at E8C13 were incubated with 1 µM WIN, 10 µM CBD, or 10 µM THC for 24 h. (**A**) Representative blot of 4 independent experiments performed in duplicate. (**B**) Blots were quantified by densitometry and expressed as the mean fraction p-eIF2α/α-tubulin ± S.E.M. (**C**) Membrane of the experiment shown in (**D**) labeled with ponceau red. (**D**) Representative blot of 3 experiments. (**E**) Blots were quantified by densitometry and data expressed as the mean ± S.E.M. (% of control). ANOVA and Bonferroni’s multiple-comparisons test were used. ** *p* < 0.01 and * *p* < 0.05 as compared to control values.

**Figure 6 brainsci-15-00291-f006:**
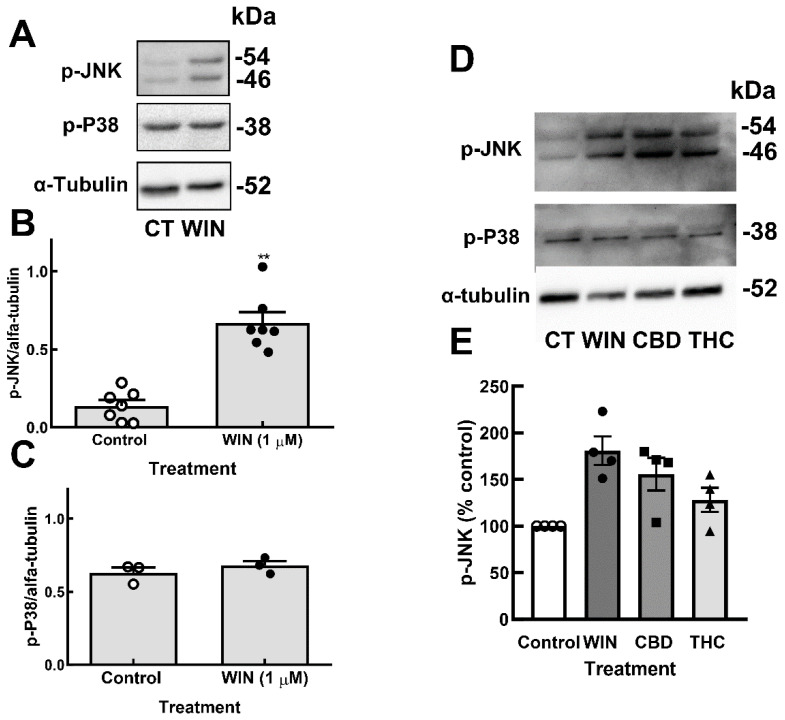
Cannabinoids induce the phosphorylation of JNK but not P38 MAP kinase in glia-enriched retinal cultures. Cultures at E8C13 were incubated with 1 µM WIN, 10 µM CBD, or 10 µM THC for 24 h. (**A**) Representative blot of the experiments quantified in (**B**,**C**) and performed in duplicate. (**B**) P-JNK. (**C**) P-P38. (**D**) Representative blot of p-JNK and p-P38 content in cultures incubated with the 3 cannabinoids. (**E**) P-JNK expression was quantified and expressed as the mean % ± S.E.M. versus the control values. ANOVA and Bonferroni’s multiple-comparisons test were used. ** *p* < 0.01 as compared to control.

**Figure 7 brainsci-15-00291-f007:**
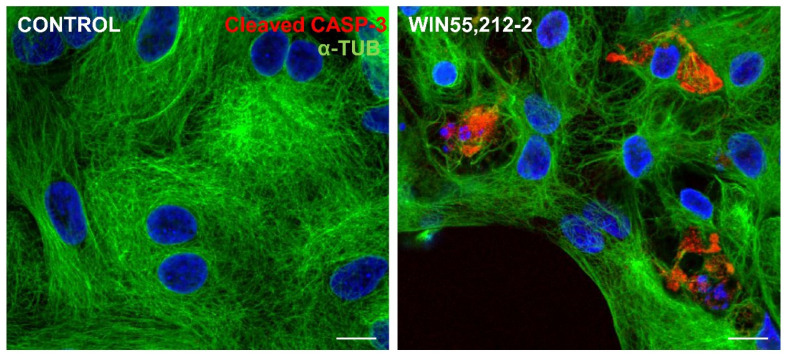
Immunofluorescence assay for cleaved caspase 3 in glia-enriched retinal cultures. Cultures at E8C13 on coverslips were treated for 24 h with 1 µM WIN, and then were fixed and immunolabeled for c-CASP 3 (red) and α-tubulin (green). Cultures were photographed on a Leica SP5 confocal microscope and the experiment was replicated 3 times, with similar results. Scale bar: 10 µm.

**Figure 8 brainsci-15-00291-f008:**
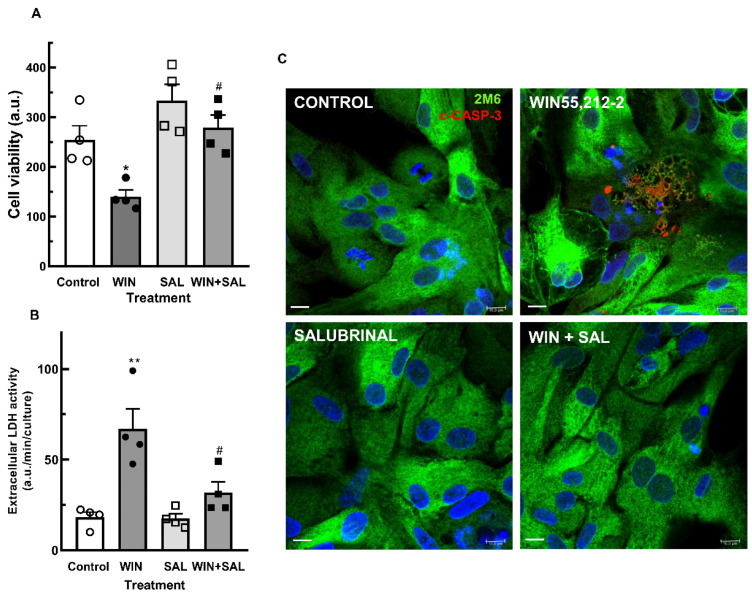
Salubrinal inhibits the decrease in cell viability induced by WIN in glia-enriched retinal cultures. Cultures at E8C13 were treated with 1 µM WIN in the absence or presence of 50 µM salubrinal. After 24 h, cultures were processed for MTT viability assays (**A**) or LDH activity (**B**). Immunofluorescence assay for c-CASP 3 in glia-enriched cultures treated with salubrinal (**C**). Cultures at E8C13 were treated with 1 µM WIN in the absence or presence of 50 µM salubrinal. After 24 h, cells were fixed and immunolabeled for c-CASP 3 (red) or 2M6 (green). Cultures were photographed on a confocal microscope and the experiment was replicated 3 times, with similar results. Data in (**A**,**B**) represent the mean ± S.E.M. of 4 experiments performed in triplicate. ANOVA and Bonferroni’s multiple-comparisons test were used. * *p* < 0.05 and ** *p* < 0.01 as compared to control. ^#^
*p* < 0.05 as compared to WIN-treated cultures. Representative micrographs are shown in (**C**). Scale bar = 10 µm.

## Data Availability

The original contributions presented in this study are included in the article/Appendix A. Further inquiries can be directed to the corresponding author.

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
