# Peer review of "Cannabinoids Activate Endoplasmic Reticulum Stress Response and Promote the Death of Avian Retinal Müller Cells in Culture"

_brainsci, 2025, doi:10.3390/brainsci15030291_

Round 1
Reviewer 1 Report
Comments and Suggestions for Authors
Thank you for the opportunity to review this manuscript and to learn about the interesting new findings. This paper describes effects of cannabinoids (WIN, CBD, THC) in Muller cells. Authors showed that cannabinoids induce the apopto-27 sis of glial cells in culture by promoting ROS production, ER stress, JNK phosphorylation and 28 caspase-3 processing. This is a very interesting study. Overall, the manuscript is very interesting and well written. Maybe some additional information should strengthen the manuscript. I believe that addressing the following points will improve the current manuscript:
Methods:
1/ How MTT results are standardized? Protein amount or directly based on the overall number of cells seeded?
Results/Discussion:
1/ Line 235: Cytoplasmic vacuolization in retinal glial cells culture are only presented for the WIN compound. According to me it is of great importance to have data from the CBD-treated conditions (CBD, THC), since it is also part of the study. Similarly, the vacuole sizing (line 236) between the different treatment should be interesting to have. It will allow to deal with difference in cellular effect. This data will reinforce the discussion especially the statement line 366-369.
2/ Figure 3: Is there any difference of cellular density? Control cells seemed to be in a lower state.
3/ Line 248 to 255: Maybe a statistical analysis should be performed on vacuolar volume to reinforce the conclusion and the discussion.
4/ Figure 3 and 8: Is there any data with CBD and THC?
5/ Figure 8: Is there any Western Blot data on cleaved caspase-3 to get an idea on the engagement apoptotic process? This data will reinforce the discussion especially the statement line 422-423.
Author Response
Please, see the attachment.

Reviewer 2 Report
Comments and Suggestions for Authors
The manuscript is dedicated to investigating the effects of non-selective cannabinoid receptor agonists, particularly WIN 55,212-2, on chicken retinal glial cells in culture. The authors demonstrated that activation of cannabinoid receptors in glial cells induces apoptosis mediated by endoplasmic reticulum stress. This work can be recommended for publication after some revisions.
Major points
- Indeed, as the authors note, the effects of cannabinoid receptor agonists on neurons and astrocytes can be diametrically opposite; specifically, in the case of neurons, cannabinoid receptor agonists act as neuroprotective, particularly antiepileptic compounds. In the case of astrocytes, cannabinoid receptor agonists, particularly WIN 55,212-2, can induce an increase in intracellular Ca2+ concentration at concentrations comparable to those used by the authors in their experiments. Although CB receptors are Gi-coupled, meaning they negatively modulate adenylate cyclase activity, similar effects of Gi-coupled receptor agonists on calcium homeostasis in glial cells, specifically astrocytes, have been demonstrated previously in scientific literature. Since the authors indicate induced endoplasmic reticulum stress, which according to the references cited in the discussion section can cause dysregulation of calcium homeostasis, assessment of changes in intracellular Ca2+ concentration in retinal glial cells upon addition of cannabinoid receptor agonists could demonstrate an additional aspect of cannabinoid action and link the authors' data with previously published findings.
- To assess the enhanced production of reactive oxygen species in retinal glial cells, the authors use DCFH2-DA, which is an indicator of general oxidative stress without significant selectivity for individual ROS, which in this case adequately meets the experimental objectives. Enhanced production of reactive oxygen species can be, in particular, a consequence of calcium overload described above, while this dysregulation can undoubtedly be observed only in specific populations of glial cells. However, this fact does not explain why in the confocal images provided by the authors, some cells are not stained with DCFH2 at all, since even basal production of reactive oxygen species (not only induced enhancement) would cause slight DCF fluorescence inside cells. The absence of fluorescence can be explained in two ways: 1) unstained cells are dead and lack active esterases capable of cleaving the diacetate tail from the DCFH2 molecule; 2) the authors performed mathematical processing of confocal images and subtracted the background, which led to the absence of even weak fluorescence. In any case, the absence of DCF fluorescence requires explanation.
- It is necessary to clearly justify the reason for using 4-PBA and its mechanism of action in the text describing the results presented in Figure 4.
Minor points
- In Figure 1, it might be better to make the y-axes and their corresponding curves red and green.
- It would be better to add the name of the statistical method used in the figure descriptions.
Author Response
Please,see the attachment

Reviewer 3 Report
Comments and Suggestions for Authors
In this original article by Marques Ventura and colleagues, the authors evaluated the effects of different cannabinoids (CBD, THC, CP55940 and WIN-55,212-2) in enriched cultures of chick retinal glial cells. In the study, the authors explored cell survival, apoptotic pathways and cellular stress. Altogether, the authors discuss the variability of the effects of different cannabinoids in different cell types.
The study is interesting and systematically addresses a gap of knowledge. However, the figures and the way that the data is presented must be improved, and the methods require more details and clarifications for reproducibility purposes. The purity of the Müller glia at each batch of cells used is ill-defined, and this can be an important pitfall in this model that has not been addressed adequately. Furthermore, I could not find experimental evidence of expression of the cannabinoid receptors studied in this specific chick model nor if there exist variability depending on the glial purity.
The fact that the study tries to study systematically different cannabinoids but does not provide data of the different experiments for all the cannabinoids is strange and inadequate. Why is that sometimes the authors decide to test specific concentrations and no other for certain tests and not others?
Importantly, the effects of cannabinoids have been found contradictory in the literature, likely due to some studies using healthy cells whereas others investigated these effects in diseased cells. Despite the study by Marques Ventura and colleagues used only healthy chick retinal cells, their rationale has been largely built on a mixed literature that is arguably confounding.
Most of the results presented in the introduction were from studies on disease models, mostly tumoral cells, which can have altered cellular responses, making the results of these studies hard to relate to these other models. There are some interesting reviews and articles that relate effects of cannabinoids on healthy cells, like these ones that would adhere better to the research presented instead:
Pagano S, Coniglio M, Valenti C, Federici MI, Lombardo G, Cianetti S, et al. Biological effects of Cannabidiol on normal human healthy cell populations: Systematic review of the literature. Biomed Pharmacother. 2020;132:110728.
Abyadeh M, Gupta V, Paulo JA, Gupta V, Chitranshi N, Godinez A, Saks D, Hasan M, Amirkhani A, McKay M, Salekdeh GH, Haynes PA, Graham SL, Mirzaei M. A Proteomic View of Cellular and Molecular Effects of Cannabis. Biomolecules. 2021 Sep 27;11(10):1411. doi: 10.3390/biom11101411. PMID: 34680044; PMCID: PMC8533448.
Additional methodologies could also improve the credibility of the results. The quantification approaches on microscopy data are poor and not too convincing. A clarification about the controls used in the study would also be helpful and strengthen the study (does 0µM refer to vehicle? What are the vehicles for each cannabinoid investigated? Particularly when comparing different cannabinoids while using the same “control”). Use of different vehicles will likely require to perform new experiments.
Considering that many different cannabinoid ligands are mentioned in the text, it could be of use to describe more precisely their effects (agonism, antagonism…) on the receptors mentioned in the text. Moreover, the authors talk about CB1R and CB2R, but they do not mention other cannabinoid receptors except TRPV1, and do not talk about other targets such as the cannabinoid enzymes. One of the ligands selected for study is CBD, which is not of high affinity to CB1 or CB2 receptors; it would be important to mention its other potential targets.
Additional contextualization of this subject in the actual literature, applications of these discoveries in the field, and how these discoveries complement what has already been discovered would make the article more convincing and interesting to read. For example, the authors mention in the introduction some interest of the field in using these cannabinoids in potential treatments for various ocular pathologies. The newfound results could be related back into that context.
Here is a list of specific constructive comments suggested for the authors:
Line 41-44: The use of “source of neuron” is unclear in this sentence, maybe use “progenitor” instead? Review the syntax of this sentence.
Line 52: The authors mention cannabinoids can activate at least 2 GPCRs. Which ones?
Lines 54-56: The authors are explaining that the receptors and enzymes are expressed in the retina, but none of the cited articles is an original article using a chick model. Pattern of expression can vary between species as shown in the cited article [10] so it is quite important to make sure that some tests are made in the chick model specifically. The authors must provide experimental evidence of expression of the receptors.
Line 56: Do you mean synthesis enzyme?
Line 69-71: Be more specific; are the cannabinoids used in this study agonists or antagonists of the main cannabinoid receptors?
Line 72 – 75: In the text, the authors mention that a non-CB1 agonist can have neuroprotective effects. However, it is also mentioned that CB1 is a main driver for pathways for neuroprotection. This needs clarification. Once again, would maybe help to introduce other targets that could potentially be responsible for these effects. You mention TRPV1 in the next paragraph: please add more details.
Line 80: What is “MethAEA”?
Line 65-92: Some articles stating neuroprotective effects are noted, and later at line 84, cytotoxic effects are noted. These paragraphs represent the two-sided effects that cannabinoids can have, but the first paragraph does not consider the effects of the ligands on their targets and seems to miss a point. Were these studies done on the same type of cells? In health or disease? How appropriate are in the context of this paper? The ligands noted in the first paragraph are agonists, while two molecules in the other paragraph are antagonists of CB1 and CB2. Therefore, the examples are not good at showing the fact that these receptors can have cytotoxic effects, it seems that the molecules are the ones responsible, and this goes against the “CB1 neuroprotective theory”. These paragraphs should be more nuanced.
Line 90: Correct “cytokine”
Line 118: When looking at the reference 26 for further details about the protocol, it can be noted that this article refers to other sources in their protocol for both mixed retinal cell and Müller cells culture. Which source is the actual original protocol used? The protocol to reproduce the technique should be as present as possible in the paper and I would encourage the authors to provide at least a short version of the method even if citing other original sources.
Line 121-124: The authors mention that the retinas are dissected, but do not mention how to ensure the separation of Müller glia from the other retinal components. How are the cultures “glial-enriched”?
Line 132: What is the culture medium used? Is it what makes these glial cells proliferate more and makes this culture “glia enriched”? Not clear.
Line 133: The cannabinoids selected and the doses at which they are used should be noted, as well as the control used (not mentioned anywhere). Also, why were these treatment times selected?
Line 139: Is this protocol for the MTT following the manufacturer instructions or elsewhere in the literature? Clarify.
Line 172: What were these antibodies diluted in? Indicate each catalog number and company precisely.
Line 173 and 180: What were the membranes rinsed with after the primary antibody?
Line 177: Were the blots stripped before the re-probing?
Line 196: As Hoechst is a non-specific nuclei marker, is there any way with this technique to make sure that this measurement of ROS is specific to glial cells? Purity is not determined.
Line 206: The term “Müller glia” is used, but are the cultures representative of this cell type only? Make sure to explain more what is a “glial cell enriched monolayer culture”, as mentioned in previous comments.
Line 208-210: Clarify the reason to choose the range of the dosages used and the times of incubation here or in the discussion (CBD stays over 24 h in the organism, for instance). Why was WIN only tested with one dose and the other cannabinoids with multiple concentrations? What are the ranges for dosing in vivo? For instance, 10uM of CBD is way above a realistic therapeutic or recreational dose found in humans, but it is relevant to explain the reason to study concentrations unlikely found after cannabis consumption.
Figure 1: Do not use AU. How do you measure cell viability? Use % or folds of increase. How is the EC50 determined? This should be expressed too.
Figure 1: It is not clear how the authors explain the variability between the viability observed with WIN in the experiments in A, B and C?
Figure 1: What are the concentrations compared to? If the control is 0, it would be beneficial to show this comparison between treatment and control more clearly and graphically.
Figure 1C: Why does CP55940 have more concentrations tested than the other substances?
Figure 2: Is there a reason why the time-lapse was only tested for CBD and WIN and not for CP55940 or THC?
Figure 2: Use color-based legends to represent the substance tested. The figures are extremely hard to follow.. Use same threshold units for the Y axis to better visualize the differences present between the 2 cannabinoids tested.
Line 235: Were the number and sizes of the vacuoles quantified with a specific technique or were these simple observations? Not clear how unbiased this is. Is this reproducible with different batches of cells?
Figure 3: Vacuoles could be indicated with arrowheads. On the control picture, what are the big clusters of cells? Some cells look like neurons; are there other pictures that might make Müller glia more visible and contain less presence of other cells?
Line 245: “Endoplasmic reticulum” repeated (acronym already presented on line 241).
Figure 4: The scale bar notation on the figure is too small and blurry. Add the name of markers on the picture.
Figure 5: The scale bar notation on the figure is too small, non-harmonized and blurry. The name of markers must be added on the picture.
Figure 5: The graph could be separated into a part B. In this graph, it would be useful to represent more graphically that the significative difference is between the treatment and the control. What is the total of cells evaluated in each experiment? It would be good if this information is present in the legends too.
Line 267: It would be useful to justify the doses selected here as not all doses were tested.
Line 283: Is there a chance that some non-glial cells might be stained by the Hoechst marker? If so, that could affect the calculations, similarly to note with comment on line 196.
Line 290: The authors should add all doses for the 3 substances tested to have consistent results to support the conclusions.
Figure 6A: Molecular weight of protein must be added to the figure. It is Alpha-Tubulin, not Tubulin. There is also some redundancy observed in the presented membrane that could be removed to make the figure simpler. The authors need to verify non-phosphorylated levels of eIF2a and do a ration to total eIF2a instead of B-Tubulin.
Figure 6B: Significative difference between THC treatment and control must be represented in a more graphical manner.
Line 302: “1μM WIN or 10 μM CBD and THC” should be replaced by “1μM WIN, 10 μM CBD or 10 μM THC” because “and” can be confusing.
Figure 7: Why is the WIN treatment shown in duplicate on this figure? Are there differences in the treatment that was done? If not, why are the measures of “ratios over tubulin” done for WIN but not for the other treatments? For the blot represented in D, some elements seem to be missing, such as the graphical representation of p-P38
Figure 7: The authors must show total levels of JNK and P38 and do ratios of the phosphorylated form to total protein, not Alpha-Tubulin.
Figure 7A: Molecular weights should be added, and the samples could be identified.
Figure 7D: This membrane has a lot of background noise, is there another picture that could represent better the tendency seen on the graph? Also, the molecular weights should be added to proteins, and the samples should be identified here too.
Figure 7E: Why is this graph “% of control” when nowhere else was presented this way? Was this data also normalized to α-tubulin?
Line 319: What criterion to avoid selection biases in CASP3 detection was used? Any positive control? It would be convenient to verify this increase of c-CASP3 with other techniques such as western blotting, or caspase-glo assay
Figure 8: The blue marker must be identified on the image.
Line 335: Would there be a way to quantify an increase in expression of cleaved CASP-3 ? (same as line 319)
Figure 9: The figure is clear but if previous ones are modified, it could be homogenized with the rest to represent statistical difference between control and treatments.
Line 350-351: It might be more relevant to present results from a healthy cell type in which some toxic effects were noted, like for example in brain neurons and astrocytes (example: Jurič DM, Bulc Rozman K, Lipnik-Štangelj M, Šuput D, Brvar M. Cytotoxic Effects of Cannabidiol on Neonatal Rat Cortical Neurons and Astrocytes: Potential Danger to Brain Development. Toxins (Basel). 2022;14(10).).
Line 352: Article [16] is a review. It could be more relevant to cite the original articles and give more specific cannabinoids and targeted populations of cells.
Line 375: There seems to be a theme surrounding tumoral cell types, therefore there could be something interesting to discuss in that sense, but in the context of this article, it could be useful to explain more how your results, observed in healthy cells relate to tumoral cells.
Line 386: Once again, this connection with tumoral cells is interesting, but more links with non-tumoral examples could give more insights into the non-pathological pathways that could be implicated.
Line 392: It would be good to mention if there are effects of THC or CBD on P38, which does not seem to have been shown in figure 7.
Line 409: are there other examples that are more closely related to healthy normal cells?
Line 411: What are the effects of JNK inhibitors, without cannabinoids, on your model? Are these effects what was observed in all the previous studies (most of which seem to have been done in tumoral contexts)? Maybe this does not work because a dephosphorylation inhibitor is not the same as a phosphorylation inductor (activation of JNK)?
Line 430: Are there other articles that suggest similar mechanisms of apoptosis caused by exposition to cannabinoids?
Line 434: I discourage the authors to use the term “retinal Müller cells” while using mixed cultures
Comments on the Quality of English Language
Syntax is confusing at times, and it would benefit from some language-proof editing.
Author Response
Please, see the attachment.

Round 2
Reviewer 1 Report
Comments and Suggestions for Authors
The authors have taken the various comments into account. The manuscript is hereby reinforced with the modifications made. No further comments. This is a very interesting study.
Author Response
We thank very much reviewer for the comments and suggestions.
Reviewer 2 Report
Comments and Suggestions for Authors
The authors have addressed all of my comments in the revised manuscript.
Author Response
We thank very much reviewer for his/her comments and suggestions.
Reviewer 3 Report
Comments and Suggestions for Authors
I thank the authors for their response and for sharing further details about the methods used. However, I still believe that my main concerns are poorly addressed and that this paper would benefit of a more in-depth reformatting.
My main concern is that the authors have neglected that the effects of cannabinoids is potentially different in health and diseased cells/tissues. To base their rationale on a plethora of articles that intertwine both is simply inaccurate and misleading. The work presented is enterily on a healthy state of the cells, yet the arguments are based on research performed in disease models. I acknowledge that the authors have included some sections detailing the mechanisms more in depth (line 416). However, this research needs to be reframed, and the rationale, subjected to a better scrutiny in the literature that differentiates health from disease.
Most of my comments regarding the figures have been largely neglected. Figures must be improved in terms of data and presentation (colors, labels, etc.).
Comments relative to the action of other cannabinoids has been largely neglected.
The authors do not justify with pertinence their choice for specific concentrations to test and of vehicle controls. Most of the doses studied are pharmacologically and physiologically irrelevant, and there is no rationale to this. Literally, the authors respond "no special reason".
When studying phosphorylation, the authors must verify total protein levels of the phosphorylated one. This would be even possible re-staining the current Western blot membranes, yet the authors decided to neglect this comment too.
Specific comments:
Comment #2 : response is a bit unclear in my opinion. And this is not explained in the article. Moreover, authors did not justify the doses used, contrary to what is mentioned in their response.
Comment #3 was generally ignored/refused, as there was no additional result from a non-pathological context that was added. I think that it is a good point to only include cell types from the retinal, but also it would be more representative to compare it to other types of glial cells such as astrocytes, which are much more like muller cells than photoreceptors for example.
Comment # 5 was also ignored/refused by authors.
RESPONSE line 70-72 okay, but still missing the action of the other cannabinoids included in this work, namely THC and CBD.
Correction in the revised version : “Moreover, both tetrahydroxycannabinol (THC) and the non-CB1 agonist cannabidiol (CBD) …” by “Moreover, both tetrahydroxycannabinol (THC) and cannabidiol (CBD)….”. (line 73 of the revised version).: This does not change the fact that the statement is the same and that it still is counterintuitive... and no more details were added.
Comment about lines 65-92 (80-92 of corrected version): Almost no changes were made. There still needs to be more nuanced clarifications on effects related to non-pathological cells. Paragraph starting at line 85 starts in a confusing manner by talking about cytotoxic effects but then jumping to a neuroprotective example. Examples of pathological cell types are mixed with non-pathological contexts, which is making it harder to understand/ find logical links.
Comment about lines 208-210 (222 of corrected version) was not answered. We are aware that the doses were not tested in vivo, the question was to understand how this related to a clinical context. 10uM is not a realistic in vivo dose, therefore it makes no sense to test this in vitro.
Comments about figure 1 were ignored.
Figure 2 : I have suggested the authors to choose combinations of colors that are visible for colorblind individuals (avoid red-green). The figures are suboptimal.
Figure 3 : identify vacuoles with arrows, as I have previously suggested.
Figure 6A : the non-phosphorylated protein still needs to be verified to make sure there is no variation. Total proteins on the gel should also be verified even if the amount of protein is equalized.
Figure 7 : normalization with alpha-tubulin might once again be biased as there is a variation in protein expression between samples.
7A and D : samples should be identified on the blots. I am not satisfied with the blots shown in 7D. Also should add the non-phosphorylated protein at the same time
Author Response
REVIEWER 3
We would like to apologize reviewer for not responding his/her comments satisfactorily in the first round. We are grateful to reviewer for the remaining criticisms and suggestions. Responses to comments are bellow.
Major comments:
- My main concern is that the authors have neglected that the effects of cannabinoids is potentially different in health and diseased cells/tissues. To base their rationale on a plethora of articles that intertwine both is simply inaccurate and misleading. The work presented is enterily on a healthy state of the cells, yet the arguments are based on research performed in disease models. I acknowledge that the authors have included some sections detailing the mechanisms more in depth (line 416). However, this research needs to be reframed, and the rationale, subjected to a better scrutiny in the literature that differentiates health from disease.
RESPONSE: We looked at many works in the literature trying to collect data on healthy cells. We described and discussed data from some of these studies performed mainly in astrocytes and neurons, as suggested. We still discussed some data from tumor cells since many studies showing a direct effect of cannabinoid inducing ER stress in healthy cells are not available. We tried to separate, in text, data from healthy cells from those obtained using disease models. Modified text in introduction, discussion and conclusion is in blue color. In the introduction section we included:
1-Information on the signal transduction of CB1 and CB2 in neurons and astrocytes;
2-information on the effect of CB1 activation in synapses in the CNS was included; 3- Neuroprotective effects of cannabinoids against excitotoxicity in healthy cells were included and separated from neuroprotective effects in retinal disease models;
4- the cytotoxic effects of cannabinoids were separated from the neuroprotective results; cytotoxic effect in hippocampal neurons was included; cytotoxic effect of THC in the retina was included;
5- then, the cytotoxic effects obtained in retinal diseases models were reported;
6- The cytotoxic evidence in neurons, astrocytes and retinal cells during development were then reported;
- Most of my comments regarding the figures have been largely neglected. Figures must be improved in terms of data and presentation (colors, labels,etc.).
RESPONSE: Figures were modified according to your suggestions and Editor’s comments;
- Comments relative to the action of other cannabinoids has been largely neglected.
RESPONSE: We are not sure that we understood this comment of reviewer and we apologize for that. We used THC and CP55940 in order to confirm previous data showing that WIN affected retinal glial cells by activating CB1 and CB2 receptors since these compounds are full agonists for these receptors; on the other hand, we know that the pharmacology of CBD is complex, activating different receptors and having other targets. However, we decided to show the results obtained with CBD as it is the most studied cannabinoid due to the absence of psychotropic actions.
- The authors do not justify with pertinence their choice for specific concentrations to test and of vehicle controls. Most of the doses studied are pharmacologically and physiologically irrelevant, and there is no rationale to this. Literally, the authors respond "no special reason".
RESPONSE: The doses used in the work were chosen for the following reasons:
- WIN – we used 1µM concentration based on the dose-response experiments performed previously by Freitas et al., 2019;
- CBD and THC- we used 10 µM concentration based on the dose-response curves described in fig.1; Since these compounds are hydrophobic, we used a concentration that provoked the maximal effect possible with the minimum effect due to their hydrophobicity. DMSO was used to dissolve these compounds and always added to the controls at the maximal concentration used in the experiment. Its concentration was kept to a minimum and adjusted to equalize DMSO concentration in all the conditions tested so that its toxic effect, if present, would be equal in all treatments. The range of concentrations used in the experiments of figure 1 were chosen based on the concentrations used by others in “in vitro” studies, mainly those using cultures. We found studies using from 0.1µM to 25 µM. Moreover, as Juric and cols (2002) reported, relevant doses of CBD that are used in humans can attain 2-3 µM. Moreover, they show that 0.1 µM or higher concentrations of CBD already induce a reduction of cell viability in cultures of neurons and astrocytes.
5- When studying phosphorylation, the authors must verify total protein levels of the phosphorylate done. This would be even possible re-staining the current Western blot membranes, yet the authors decided to neglect this comment too.
RESPONSE: Unfortunately, we cannot perform these experiments, as we don’t have these membranes anymore; We apologize reviewer but ask him/her to take our data into account. As reviewer knows, the regulation of MAP kinases usually occurs by phosphorylation and not by modulation of protein expression, since these proteins are “hubs” in intracellular signaling. Moreover, if protein kinase levels decreased due to the treatment, the control, basal phosphorylation would provide a false increase in the phosphorylation of the protein. In addition, a decrease in protein levels of JNK and p-38 that indicated a false positive effect on their phosphorylation could be obtained if cells died in the cultures. To avoid this, we added the same protein levels to gels. We stained membranes with Ponceau red to determine the transference of proteins to membranes and used anti-alfa-tubulin to ascertain that the gel loading and ECL detection were constant among samples. We used the fraction phospho-protein/alfa-tubulin only to correct small variations in protein content that usually occurs between samples.
Specific comments:
Comment #2: response is a bit unclear in my opinion. And this is not explained in the article. Moreover, authors did not justify the doses used, contrary to what is mentioned in their response.
RESPONSE: Please, see the response to the major comment number 4 above. Moreover, based on the dose-response curves, we used a concentration that provoked the maximal effect possible with the minimum effect due to their hydrophobicity that we could detect. 1µM WIN and 10 µM THC or CBD were used for all the subsequent experiments on ROS detection, western blotting etc.
Comment #3 was generally ignored/refused, as there was no additional result from a non-pathological context that was added. I think that it is a good point to only include cell types from the retinal, but also it would be more representative to compare it to other types of glial cells such as astrocytes, which are much more like muller cells than photoreceptors for example.
RESPONSE: Please, see the detailed explanation to the comment 1 of the major points; we modified extensively the text of the introduction and the discussion to include data obtained in healthy cell types.
Comment # 5 was also ignored/refused by authors.
RESPONSE: We apologize reviewer and as a first attempt to explain why we did not describe the pharmacology of the compounds we used in the work, we ask him/her to look at the response for comment 3 of the major points; As we and reviewer know, the pharmacology of cannabinoids is extremely complex, manly when looking at CBD; In the first moment, our goal was to investigate if these compounds affected cell viability (and investigate some of the mechanisms involved) in retinal glial cultures. However, to determine which subtype of receptor or which protein is affected by them is a relevant issue. We don’t have these data now. Thus, we respectfully ask reviewer not to consider this as a refuse to his/her suggestion and to exclude these pharmacological data. There are too many extensive reviews reporting contradictory results that we hope reviewer agrees that reporting pharmacological aspects and targets without evidence can be more confusing.
RESPONSE line 70-72 okay, but still missing the action of the other cannabinoids included in this work, namely THC and CBD.
RESPONSE: We included the reference [29] in lines 102-104 of the revised manuscript to include information on THC effects on the retina. We did not find data on the effect of CP55940 in the retina. The effects of CBD on the retina and on cultures of astrocytes and neurons were included in the introduction and discussed in the discussion section. Please, see text in blue in the introduction and discussion.
Correction in the revised version: “Moreover, both tetrahydroxycannabinol (THC) and the non-CB1agonist cannabidiol (CBD) …” by “Moreover, both tetrahydroxycannabinol (THC) and cannabidiol (CBD)….”. (line 73 of the revised version).: This does not change the fact that the statement is the same and that it still is counter intuitive... and no more details were added.
RESPONSE: We are not sure that we understood this comment of reviewer and we apologize for that. We changed this sentence to “Moreover, both tetrahydroxycannabinol (THC) and cannabidiol (CBD), as well as the synthetic cannabinoid WIN55,212-2, protect the retina against NMDA-induced excitotoxicity [23,24]”. Unfortunately, we did not understand what is counterintuitive. We apologize.
Comment about lines 65-92 (80-92 of corrected version): Almost no changes were made. There still needs to be more nuanced clarifications on effects related to non-pathological cells. Paragraph starting at line 85 starts in a confusing manner by talking about cytotoxic effects but then jumping toa neuroprotective example. Examples of pathological cell types are mixed with non-pathological contexts, which is making it harder to understand/ find logical links.
RESPONSE: We made several modifications in the Introduction section; please, see the response do comment 1 of the major comments; Please, see text in blue in the introduction section.
Comment about lines 208-210 (222 of corrected version) was not answered. We are aware that the doses were not tested in vivo, the question was to understand how this related to a clinical context. 10 uM is not a realistic in vivo dose, therefore it makes no sense to test this in vitro.
RESPONSE: In order to better answer this criticism, we decided to compare the doses that are used in anti-convulsant therapy and the dose used in our study. We calculated for cannabidiol since this is the most used compound. Please, follow the calculations below:
Regarding the dose of cannabidiol:
According to the article “A systematic review of cannabidiol dosing in clinical populations”. by Millar SA, Stone NL, Bellman ZD, Yates AS, England TJ, O’Sullivan SE. -Br J Clin Pharmacol. 2019 Sep;85(9):1888-1900. doi: 10.1111/bcp.14038,
the collected studies used cannabidiol doses ranging from 1 to 50 mg/kg/day.
So, if an 80kg patient uses 50mg/kg/day, in one day this individual will have ingested 4000 mg (4g) of CBD per day.
Thus, what would be the plasma concentration of CBD in this patient?
Step1: Convert the CBD dose from grams to micrograms:
CBD dose: 4 g
Convert to micrograms: 4 g x 1,000,000 μg/g = 4,000,000 μg
Step 2: Calculate the amount of CBD absorbed:
By using an absorption rate of ~19% (or 0.19) described in “Frontiers in Pharmacology, v.9, 2018, doi: 10.3389/fphar.2018.01365”, the amount of CBD absorbed would be 4,000,000 μg x 0.19 = 760,000 μg
Step 3: Calculate the volume of distribution of CBD
By using the volume of distribution of 32.7 L/kg described in “doi: 10.3389/fphar.2018.01365” and a body weight of 80 kg, the total volume of distribution would be= 32.7 L/kg x 80 kg = 261.6 L
Step 4: Calculate the plasma concentration of CBD
Plasma concentration (μg/mL) = Amount of CBD absorbed / Total volume of distribution;
plasma concentration (μg/mL) = 760,000 μg / 261,600 mL ≈ 2.9 μg/mL
Therefore, the plasma concentration of CBD in an 80 kg patient using 4g of CBD may be approximately 2.9 μg/mL.
In the retinal cultures:
how many micrograms of cannabidiol per ml we have in the cultures. If I chose the concentration of 5 μM of CBD to do the calculations.
The molar mass of CBD is approximately 314.47 g/mol.
If we consider a volume of 1 mL, we can calculate the amount of CBD in μg as follows:
Amount of CBD (μg) = Concentration (μM) x Molar mass (g/mol) x Volume (L) x 1000
Substituting the values:
Amount of CBD (μg) = 5 μM x 314.47 g/mol x 0.001 L x 1000
Amount of CBD (μg) ≈ 1572.35 μg
Therefore, in a 5 μM cannabidiol solution, there are approximately 1572.35 μg of CBD per liter.
CBD in μg per mL, divide the value above by 1000 (since 1 L = 1000 mL):
Amount of CBD (μg/mL) ≈ 1572.35 μg/L ÷ 1000 = 1.57 μg/mL
if one uses a 10 µM solution = the concentration will be 3,14 μg/mL
Conclusion:
To have an anticonvulsant effect in patients, a plasma concentration of ~2.9 μg/ml is necessary. In order to kill retinal cells in culture, a concentration of ~ 3.14 μg/ml is necessary. Therefore, we believe that we are working within or close to the therapeutic doses of CBD commonly used.
Comments about figure 1 were ignored.
RESPONSE: We apologize for that. We tried to answer the questionings. However, figure was modified following editor’s request.
Figure 2: I have suggested the authors to choose combinations of colors that are visible for color blind individuals (avoid red-green). The figures are suboptimal.
RESPONSE: Reviewer 2 asked us to change curves to red and green; We did not know that data should be visible for color blind individuals; For them, symbols and line types are different (open squares, circles, filled squares etc.).
Figure 3 : identify vacuoles with arrows, as I have previously suggested.
RESPONSE: done.
Figure 6A : the non-phosphorylated protein still needs to be verified to make sure there is no variation. Total proteins on the gel should also be verified even if the amount of protein is equalized.
RESPONSE: Please, see response to the major comment 5. We always determine protein content in the samples by the Bradford method; New standard curves are always performed in each determination.
Figure 7 : normalization with alpha-tubulin might once again be biased as there is a is a variation in protein expression between samples.
RESPONSE: Please, see response to the major comment 5. There was very little variations in alpha-tubulin expression. Protein content in each sample was determined in triplicate.
7A and D : samples should be identified on the blots. I am not satisfied with the blots shown in 7D. Also should add the non-phosphorylated protein at the same time
RESPONSE: Done.

Round 3
Reviewer 3 Report
Comments and Suggestions for Authors
I thank the authors for their detailed response and for considering some of my comments.
I can accept that, in the revised version, the authors contextualize better when the references related to a health or diseased context. However, the authors fail to justify the doses employed.
As per the arguments used by the authors and the cited papers, consuming (a lot) of cannabis leads to concentrations of CBD 3µM, whereas all experiments are CBD 10µM (for instance). In my opinion, this is not in the same range as to have a clinically relevant impact. For instance, regard Figure 1 viability dosing: doses of CBD at the range of 3µM are subtoxic whereas at 10µM halve the cell population. The calculations are also not pertinent, as levels of CBD/THC in the vitreous cannot be calculated as such, and the pharmacokinetics should be tested experimentally considering other parameters. Overall, current datasets with 10µM would benefit of additional experiments at 3µM (published side-by-side) and would have more relevance to the readership given their translationality.
I understand the problems of the authors to present total target protein levels in their phosphorylation studies. Yet, the pictures of total gel proteins (with Red Ponceau) are not shown in the revised version. Please add these to the main figure under each panel studying phosphorylation. That said, not conserving Western blot membranes (physically) is not a good laboratory practice, and can lead to problems and investigations related to fabricating data (as it has happened in many publications before where reprobing is needed). For the sake of the authors', I would strongly encourage them to preserve future membranes a few years after publishing their work.
Author Response
Responses to reviewer 3:
1- I can accept that, in the revised version, the authors contextualize better when the references related to a health or diseased context. However, the authors fail to justify the doses employed. As per the arguments used by the authors and the cited papers, consuming (a lot) of cannabis leads to concentrations of CBD 3µM, whereas all experiments are CBD 10µM (for instance). In my opinion, this is not in the same range as to have a clinically relevant impact. For instance, regard Figure 1 viability dosing: doses of CBD at the range of 3µM are subtoxic whereas at 10µM halve the cell population. The calculations are also not pertinent, as levels of CBD/THC in the vitreous cannot be calculated as such, and the pharmacokinetics should be tested experimentally considering other parameters. Overall, current datasets with 10µM would benefit of additional experiments at 3µM (published side-by-side) and would have more relevance to the readership given their translationality.
RESPONSE: We included a new figure 2 in manuscript illustrating the effects of CB1 and CB2 receptor antagonists on the cannabinoid-induced decrease in cell viability. As shown, the CB2 receptor antagonist SR144528 completely prevented the CBD- induced decrease in cell viability. Since CBD is a negative allosteric modulator (NAM) at CB1 receptors and a partial agonist at CB2 receptors, these results suggest that CBD induces glial cell death through the CB2 receptor activation. More importantly, these findings rule out the possibility that CBD induced cell death was due to drug toxicity resulting from a 10 µM concentration of CBD.
As suggested by reviewer, we conducted experiments to assess the effect of 3 µM CBD (alongside 10 µM CBD) on the phosphorylation of eIF2α, an early event in unfolded protein response (UPR) that follows ER stress. We included data in the new figure 5. The old combined figure showing the results of p-eIF2alfa + p-JNK was separated into new figures 5 and 6. As shown in the new figure 5, CBD at a 3 µM concentration also increases eIF2α phosphorylation, suggesting that CBD-induced ER stress occurs at lower than 10 µM concentrations of CBD. This data also supports the idea that CBD-induced cell death is not due to drug toxicity.
2- I understand the problems of the authors to present total target protein levels in their phosphorylation studies. Yet, the pictures of total gel proteins (with Red Ponceau) are not shown in the revised version. Please add these to the main figure under each panel studying phosphorylation. That said, not conserving Western blot membranes (physically) is not a good laboratory practice, and can lead to problems and investigations related to fabricating data (as it has happened in many publications before where reprobing is needed). For the sake of the authors', I would strongly encourage them to preserve future membranes a few years after publishing their work.
RESPONSE: We do not fabricate data.

Round 4
Reviewer 3 Report
Comments and Suggestions for Authors
I never stated nor suggested that the authors fabricated any data. There is a difference from fabricating data to bad laboratory practices, and my comment was a friendly suggestion for their future work. It is interesting how the authors reflected over the comment.
I am not convinced with one single experiment using 3µM. All experiments (no exception) must be done with that dose or in the range of nM (which is more physiologically relevant and more recurrent in cannabis consumers).